# Muscle Strength Training and Monitoring Device Based on Triboelectric Nanogenerator for Knee Joint Surgery

**DOI:** 10.3390/mi16121387

**Published:** 2025-12-06

**Authors:** Jing Liu, Yi Zhang, Xia Liu, Chenming Sun, Youquan Wang

**Affiliations:** 1Faculty of Rehabilitation Medicine, Jining Medical University, Jining 272067, China; fight8966@163.com (X.L.); chenming_sun1998@163.com (C.S.); jnyxykfgc@163.com (Y.W.); 2Faculty of Electrical and Electronic Engineering, Changchun University of Technology, Changchun 130012, China; 13654355660@163.com

**Keywords:** triboelectric nanogenerator, self-sensing, knee joint operation, muscle strength training, self-rehabilitation management

## Abstract

At present, there are some devices for muscle strength training after knee surgery, such as elastic bands and isokinetic muscle strength training instruments, but most of them are expensive or cannot monitor training progress. Triboelectric nanogenerators (TENGs) have proven to be reliable self-sensing devices. There have been some applications in the field of rehabilitation, but few have been used for muscle strength training. Our team has innovatively applied the TENG self-sensing device to the self-rehabilitation management of the knee joint post-surgery. We have developed the “Triboelectric Nanogenerator for Muscle Strength Training of Knee Joint after Surgery” (MSTKJS-TENG), which is significantly more integrated than traditional instruments (volume: 120 mm × 100 mm × 100 mm) and can real-time track the number and quality of movements completed by patients during muscle strength training. The development of this device has made up for the deficiencies of traditional instruments. It can assist medical staff in remotely evaluating the recovery of patients’ postoperative muscle strength to a certain extent, thereby adjusting training intensity in a timely manner and providing personalized guidance. Meanwhile, the research on this device provides effective technical support and innovation for the development of smart rehabilitation medicine.

## 1. Introduction

Osteoarthritis of the knee (KOA) is the most common skeletal joint disease among elderly populations and is also the leading cause of chronic disability in older adults [1]. At present, total knee arthroplasty (TKA) is considered the optimal treatment method for severe KOA cases. However, complications such as postoperative pain, stiffness, and functional disorders of the affected joints are prevalent, emphasizing the growing need for postoperative rehabilitation [2,3]. Muscular injury following surgery may negatively impact surgical outcomes and implant longevity, and functional muscle recovery has become increasingly important in meeting the future needs of patients [4,5,6]. Rehabilitation training is an effective means of regaining muscle strength, and patients often require prolonged periods of training following surgery. However, the current clinical rehabilitation medical resources are scarce, and self-rehabilitation management outside the hospital has gradually become an essential part of the rehabilitation process for patients. As their muscle strength improves, rehabilitation devices play a critical role in training, but current muscle training devices, which are used for self-rehabilitation management, tend to be bulky, expensive [7,8,9], or few can accurately monitor the quantity and the quality of exercises performed by patients, presenting significant drawbacks [10]. This limitation undoubtedly restricts where post-surgery patients can perform rehabilitation exercises, hindering their rehabilitation progress to some extent and potentially exposing them to risks associated with secondary surgeries. Therefore, it is essential to develop a rehabilitation device that is self-sensing and capable of assisting with postoperative knee muscle strength training while simultaneously monitoring training effectiveness.

The triboelectric nanogenerator (TENG) is an energy-harvesting device based on contact electrification and electrostatic induction, invented by Wang Zhong Lin’s team in 2012 [11,12,13]. It demonstrates remarkable characteristics for self-sensing applications, including low cost, light weight, and high energy-harvesting efficiency [14,15]. In recent years, TENG has shown great potential in various human health applications. For example, flexible TENG sensors can be used for real-time detection of gait parameters during walking, enabling comprehensive gait analysis for patients with lower-limb movement disorders [16,17]. But now, few focus on its practical applications involving TENG in post-knee-surgery muscle-strengthening exercises, especially in the self-rehabilitation management process [18,19,20,21]. Therefore, it is necessary to explore the feasibility of using TENG as a self-sensing device to monitor the effect of muscle strength training.

In this study, our team modified existing rehabilitation equipment by incorporating TENG technology, transforming the signals to enable monitoring of the quantity and quality of exercise movement completion during patient muscle-strengthening exercises. This can help patients undergo effective rehabilitation training and allows medical personnel to promptly assess the patient’s muscular recovery status. In designing and developing MSTKJS-TENG, our team tested the device under steady-state operation and measured a peak-to-peak open-circuit voltage of up to 180 V, a maximum short-circuit current reaching 15 μA, and stable sensing signals with a peak power output of up to 170 μW. Since TENG generates sinusoidal alternating electrical signals during energy generation, which are not conducive to recognition and processing by Microprogrammed Control Units (MCUs), our team designed a Signal Processing Module (SPM). This module converts sinusoidal alternating electrical signals into square-wave direct current signals, thereby improving MCU recognition rates to some extent [22,23,24,25,26].

## 2. Experimental

### 2.1. Production of MSTKJS-TENG

The overall structure of MSTKJS-TENG was designed using SolidWorks 2023 software and fabricated via a 3D printer. The main shell is made of acrylic board, which was cut by a laser cutting machine. Its external dimensions are 120 mm (length) × 100 mm (width) × 100 mm (height). The copper electrodes, evenly distributed on the acrylic board, have a thickness of 1 mm, and each electrode is of equal size, facilitating stable signal output. The copper electrodes are bonded to the acrylic board with epoxy resin adhesive. The fluorinated ethylene-propylene (FEP) film has a thickness of 0.08 mm and is attached to the rotor using an epoxy resin adhesive. The one-way bearing has dimensions of 8 mm (inner diameter) × 12 mm (outer diameter) × 12 mm (height); it is placed at the center of the TENG rotor and connected to the pull plate. The internal structure of the pull plate comprises a spring, a rotor wheel, and a central shaft. Additionally, this device is equipped with a preload structure for adjusting motion resistance, consisting of a threaded rod (M4 × 50 mm), a spring (outer diameter 8 mm, free length 20 mm), and a friction plate (polytetrafluoroethylene, diameter 10 mm).

### 2.2. Experimental Equipment

The equipment used to measure and acquire signals from MSTKJS-TENG included a programmable electrometer (model 6514, Keithley, Cleveland, OH, USA), a data acquisition card (USB-6356 BNC, Austin, TX, USA), and a rotary motor (Model 142U3E300CACAA165240, Newton, Wales, UK) providing external excitation. The 142U3E300CACAA165240 motor is an AC-excited servo motor with a maximum torque of 45 Nm, a maximum speed of 3000 rpm, and a rated current of 5 A to 15 A. It is equipped with a brake. Signals collected during the experiment were all recorded using LabVIEW 2019 software.

## 3. Results and Discussion

### 3.1. Structural Design and Working Principle

The quadriceps femoris serves as the prime mover for knee extension and often experiences noticeable decline following knee surgery, making it an essential aspect of rehabilitation training [27]. During active quadriceps-strengthening exercises, patients begin in a sitting position with hips flexed to 90° and knees bent. They then extend the knee through the full range of motion against resistance. Figure 1(a_1_) shows MSTKJS-TENG’s placement position when operational. Figure 1(a_2_) and Appendix A show a 3D rendered front view of MSTKJS-TENG. The overall structure and pre-tightening structure of MSTKJS-TENG are shown in Figure 1(b_1_) and Appendix A, respectively. This device is equipped with a pre-tightening mechanism to adjust the motion difficulty. This structure consists of a threaded rod, a spring, and a friction plate. By rotating the threaded rod, the friction plate and the spring generate an extrusion force, thereby increasing the resistance of the spring during its elastic deformation, and thus changing its resistance during knee extension. When performing the “extension” action, the pulling rope drives the spring to undergo elastic deformation, which in turn drives the rotor to rotate, thereby providing torque to the TENG rotor. When completing the “flexion” action, the spring returns to its original state and pulls the rope back into the pull disc. Since the TENG rotor uses a one-way bearing, when the pull disc returns to its deformed state, the TENG will not be affected by the counterforce. Figure 1(b_2_) displays photographs of the pulling plate and spring arrangements within MSTKJS-TENG. The torsion spring provides a restoring torque approximately proportional to its angle of rotation, but its operation in a pull plate is closer to a constant force. For a classic planar torsion spring, its torque can be approximately expressed as Equation (1):
(1)$$M\approx \frac{Ebh^{3}}{12L}$$

*E* is the elastic modulus of the helical spring material, *b* is the width of the spring, *h* is the thickness of the spring, and *L* is the effective working length of the spring.

The elastic potential energy of a coil spring can be calculated as Equation (2):
(2)U=∫Mdθ

Since the torque does not vary much within the working range of the winch, it can be simplified as Equation (3):
(3)U≈Mθ

Here, *θ* represents the total rotational angle of the coil spring when the cord is pulled. The relationship between the length *S* pulled by the cord and *θ* is as shown in Equation (4):
(4)S=θr where *r* is the radius of the spool around which the rope is wound [28].

Figure 1(b_2_) presents rotor and FEP film placements within MSTKJS-TENG. Explosion diagram C shows the internal composition and constituent materials of MSTKJS-TENG. As shown in explosion diagram C, the MSTKJS-TENG housing is made of acrylic material. The electrodes are copper with a thickness of 1 mm, arranged in differential electrode pairs; rotors, connecting rods, and bases are constructed from polylactic acid via 3D printing. In addition, there are five FEP-based tabs present.

Patients perform repetitive knee extension movements during strength-training exercises, connected to the MSTKJS-TENG by a cord. To ensure stable power generation by MSTKJS-TENG’s generating unit and accurate record-keeping during training, the research group employed a coiled spring mechanism composed of iron to guarantee cable movement upon knee extension. The device uses a spiral spring to provide torque for the TENG rotor. Mounted in a spiral spring groove with one end connected to the central shaft (Figure 1c), the spiral spring drives the rotor to rotate and generate resistance when the patient pulls the drawstring. During the patient’s return movement, the elastically deformed spiral spring reverts to its original state, pulling the drawstring back into the device to complete one pulling cycle. Since the mechanical properties of the coil spring enable the component to be automatically returned to its original position after the patient completes a standard action and provide a sustained greater resilience in a narrow space [29], the rope can be pulled back into the box to prepare for the next trip. As knee extension actions involve repeated movements and feature coiled spring mechanisms, center rotor reversals may occur. In response to this situation, the team incorporated a unidirectional bearing design where rotation occurs freely in one direction but locks in another. The above details describe MSTKJS-TENG’s structural composition and motion characteristics. Since the MSTKJS-TENG is equipped with five pairs of electrodes and five FEP film paddles, it generates five cycles of sine waves after rotating one full circle. However, the backend MCU module responsible for counting cannot identify sine wave signals. Therefore, the electrical signals generated by the MSTKJS-TENG need to be transmitted to the backend SPM for processing via wires, where they are converted into square waves. Finally, the square waves are recognized and processed by the backend MCU to achieve the counting function (detailed information about the MCU algorithm can be found in the Appendix A).

Figure 2 illustrates enlarged images of MSTKJS-TENG’s copper electrodes and FEP tabs alongside its corresponding electrical generation principles. When MSTKJS-TENG reaches the starting state as displayed in Figure 2(a_1_), FEP contacts Copper Electrode II. Based on the electronegativity order list, FEP possesses greater electronegativity than copper, causing the former to acquire a negative charge and the latter a positive charge upon contact. As the membrane moves to its position in Figure 2(a_2_), it lies between Copper Electrodes I and II, with a constant overall quantity of negatively charged particles remaining on the surface, resulting in some positively charged particles moving onto Copper Electrode I. Due to charge displacement leading to current flow, as the FEP membrane moves towards its position shown in Figure 2(a_3_), full contact occurs between it and Copper Electrode I. Maintaining the same principle, as the net quantity of negative charges on the FEP membrane stays constant while still generating consistent current, the membrane again contacts Copper Electrode II when transitioning to its position, as seen in Figure 2(a_4_). When arranging the electrodes as differential electrode pairs, designers allow opposing current directions between Figure 2(a_2_,a_3_) and later phases of contact with Copper Electrode II upon membrane repositioning. Figure 2(a_1_–a_4_) illustrates one complete cycle of motion within MSTKJS-TENG’s energy-generating unit, generating a sinusoidal alternating voltage signal useful for subsequent sensing purposes. COMSOL Multiphysics 6.0 software simulation results presented in Figure 2(b_1_–b_3_) demonstrate potential differences in contact positions between the two materials, providing effective simulation support for theoretical analysis during later stages of development.

### 3.2. Power Generation Performance Test

To systematically test TENG’s energy output capabilities, our research team utilized motorized rotations to simulate MSTKJS-TENG performance across various scenarios, thus thoroughly understanding operational reliability under varying conditions and conducting prolonged endurance testing to verify system stability over extended durations.

Figure 3(a_1_) displays peak-to-peak open-circuit voltage values for telecommunication signals produced at different rotational speeds for MSTKJS-TENG during testing. During experiments, our team selected rotation rates of 50, 100, 150, 200, 250, and 300 rpm for systematic testing procedures. Data reveals that since the contact area between the copper electrodes and FEP film remains constant, the short-circuit peak-to-peak voltage outputs remain unchanged regardless of RPMs applied. Moreover, as RPM increases, the output sine voltage signal frequency within a 0.5 s interval also rises consistently while the period decreases simultaneously. This trend persists throughout RPM range variations. As depicted in Figure 3(a_2_), MSTKJS-TENG shows increasing wave numbers with increasing RPM, indicating higher short-circuit telecommunication signal current outputs across all tested rates. MSTKJS-TENG’s inherent qualities, combined with changes in speed, result in rising current levels over time, as demonstrated by an increasing number of wave peaks per second as RPM increases. As illustrated in Figure 3(a_3_), a direct correlation is observed between the number of charge waves emitted and the rotational speed. Additionally, due to MSTKJS-TENG’s inherent traits and RPM fluctuations, the external charge density remains constant over time despite increased RPM. Figure 3(a_4_) demonstrates how adjusting load resistor values directly affects system performance when connected to the device. When load resistors are too small, impedance mismatches hinder the transfer of maximum voltage from the TENG to external loads. Consequently, during low-load resistor conditions, MSTKJS-TENG exhibits brief instances of short-circuiting, resulting in maximum current output and minimum voltage, which register near zero power output, as expressed by Equation (5).
(5)$$P={I_{R}}^{2}R=\frac{RE^{2}}{{\left(R+r\right)}^{2}}=\frac{RE^{2}}{\left({\left(R-r\right)}^{2}+4Rr\right)}=\frac{E^{2}}{\left({\left(R-r\right)}^{2}/R+4r\right)}$$

In equation form, *E* represents MSTKJS-TENG’s open-circuit voltage while r denotes the device’s internal resistance. The relationship for matching load resistance becomes evident when stated as follows:
(6)$$P_{MAX}=\frac{E^{2}}{4r}$$

Equations (5) and (6) indicate optimal power delivery via MSTKJS-TENG at external resistances of 10^7^ ohms, confirmed where voltage/current curves intersect. As illustrated in Figure 3(b_1_), we observe full-wave rectification of MSTKJS-TENG sinusoidal AC signals into DC signals, charging electrolytic capacitors post-stage. Upon each positive AC sine wave passage through full-wave rectifier circuits, diodes D_1_ and D_4_ conduct while D_2_ and D_3_ cease functioning due to unidirectional conduction properties; conversely, during periods of negative signals, D_2_ and D_3_ become active while D_1_ and D_4_ stop, completing a single cycle producing a sinuous AC waveform that proceeds to charge C capacitors downstream. As shown in Figure 3(b_2_), when the rotational speed was stabilized at 50 rpm, and all test capacitors were set to charge to 10 V, the charging rates of capacitors with different capacities showed significant differences. Among them, the 0.1 μF capacitor had the fastest charging rate, completing the charging process from the initial state to 10 V in just 1 s; the capacitor with a capacity of 0.22 μF had a slightly lower charging efficiency, reaching 10 V voltage in only 5 s; the capacitor with a capacity of 1.0 μF had a further increased charging time, taking 10 s to charge to 10 V; while the capacitor with a capacity of 2.2 μF had the slowest charging rate, requiring 50 s from the start of charging to reach 10 V. In the subsequent experiments, multiple sets of experiments were conducted at different rotational speeds, and the output performance remained stable (Appendix A).

### 3.3. Microprogrammed Control Unit Performance Test

To achieve accurate signal recognition by Microprogrammed Control Units, our team developed square-wave conversion circuits known as Signal Processing Modules. As shown in Figure 4(a_1_), these modules offer high stability, precision, adjustable duty cycles, and controllable periods, making them suitable for versatile applications [21]. Given their digital nature, square-wave conversions enhance compatibility with digital systems that require precise control. To fulfill these requirements, we opted for LM358 dual amplifiers following a comprehensive review. This model encompasses two independent, high-gain operational amplifiers with built-in frequency compensation, ideal for broad supply-voltage ranges functioning in either single or double power modes. By connecting a positive 3.3 V DC source to pin #8, grounding at pin #4, and attaching TENG output terminals to pins #2 and #3, respectively, our setup ensures adequate chip power. Resistor R1 provides filtering/buck-voltage functionality, while R_2_ serves as a pull-up grounded via common connections. Pin #1 connects directly to the MCU input terminals, with resistors R_3_/R_4_ providing impedance matching and additional pull-up functionality; the Signal Processing Modules perform square-wave transformation upon concurrent groundings between the MCU and IC chips. Post-MSTKJS-TENG testing with rotating motors, results displayed in Figure 4(a_2_), reveal square-wave period reductions corresponding to increasing RPM; maintaining constant DC power sources results in uniform square-wave voltage trends throughout these adjustments. Results verify consistent, stable MSTKJS-TENG performance throughout prolonged operation intervals.

### 3.4. Demonstration

To validate MSTKJS-TENG feasibility, we conducted verifications by replicating knee-surgery recovery scenarios in seated volunteers undergoing quadriceps strength training. Participants fastened MSTKJS-TENG straps below affected ankle joints before securing devices beneath seating arrangements. Before the training, professionals need to set different counting standards for patients with different limb lengths. The counting standard is determined by the number of square waves generated by the instrument during the patient’s standard knee extension movement. This number is set to one counting unit (counting as 1), as shown in Figure 5(a_2_). Ongoing exercises culminate in a display showing total exercise counts on MCU screens, as shown in Figure 5(a_2_). Insufficient execution during training fails to generate the required signal counts, prompting no exercise to count towards totals; fostering patient awareness/quality control. Based on individual strength thresholds, tension adjustments fine-tune resistance levels to match difficulty settings. Referencing Figure 5(b_1_), Human movement initiates cable retraction, generating torsion-induced rotor rotation within MSTKJS-TENG units, which transmit energy via square-wave conversions facilitated by powered Signal Processing Modules/Microprogrammed Control Units to analog-to-digital conversions, displaying exercise status updates on screens for seamless rehabilitation monitoring. Figure 5(c_1_–c_3_) illustrates open-circuit voltages, short-circuit currents, and charge curves for standard knee extensions, showcasing MSTKJS-TENG operations followed by digitally processed square waveforms via SPMs. These demonstrations solidify MSTKJS-TENG’s potential contributions towards autonomous sensor technologies revolutionizing intelligent healthcare solutions.

## 4. Conclusions

In summary, we demonstrate, for the first time, the integration of TENG into postoperative knee joint rehabilitation training to improve muscle strength in patients. Our muscle strength training after knee joint surgery, using a triboelectric nanogenerator (MSTJS-TENG), can monitor the standard performance of movement completion during self-rehabilitation and eliminate the need for counting. This device significantly improves the efficacy and convenience of muscle-strengthening training during self-rehabilitation management for patients, demonstrating clear advantages in practical applications. Furthermore, this system offers versatility and can be applied to muscle strength training for different patients and muscle groups, such as shoulder or hip muscles. The implementation of this technology provides a more efficient and reliable solution for the development and promotion of intelligent rehabilitation medical systems.

## Figures and Tables

**Figure 1 micromachines-16-01387-f001:**
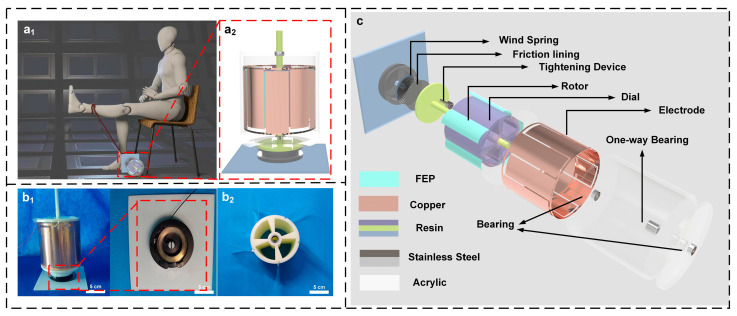
(**a_1_**) MSTKJS-TENG in use, (**a_2_**) frontal 3D view of MSTKJS-TENG, (**b_1_**,**b_2_**) photographs of MSTKJS-TENG, and (**c**) exploded 3D rendering of MSTKJS-TENG.

**Figure 2 micromachines-16-01387-f002:**
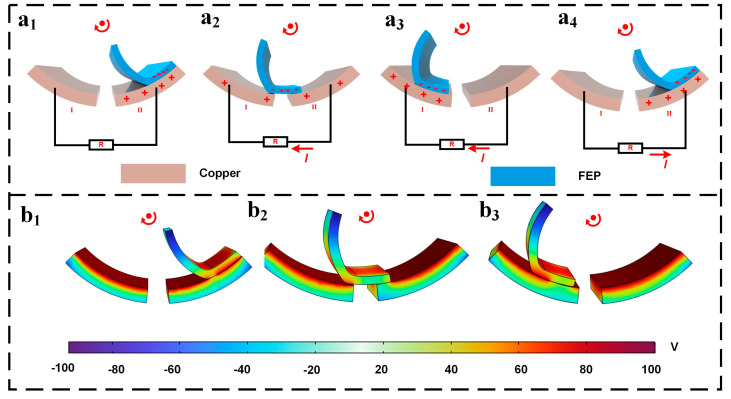
(**a_1_**–**a_4_**) Illustration of MSTKJS-TENG working principles, and (**b_1_**–**b_3_**) simulation results of MSTKJS-TENG generation unit.

**Figure 3 micromachines-16-01387-f003:**
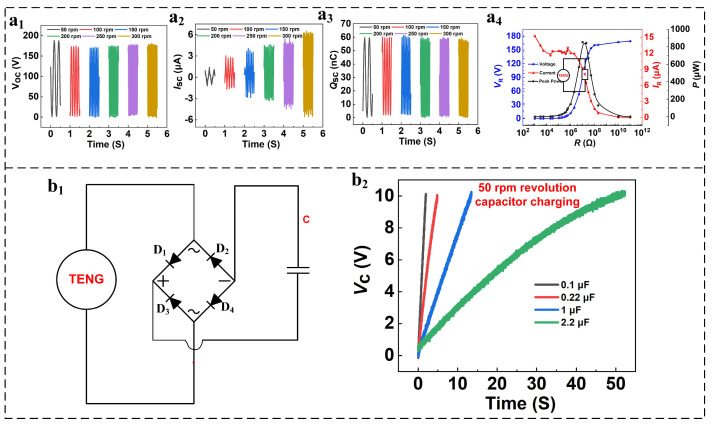
(**a_1_**–**a_3_**) Voltage, current, and charge waveforms across varied RPM ranges, (**a_4_**) power/voltage/current trends, (**b_1_**) circuit diagram depicting capacitor charging process, and (**b_2_**) voltage profiles demonstrating recharge patterns among 50 RPM/Capacitance configurations.

**Figure 4 micromachines-16-01387-f004:**
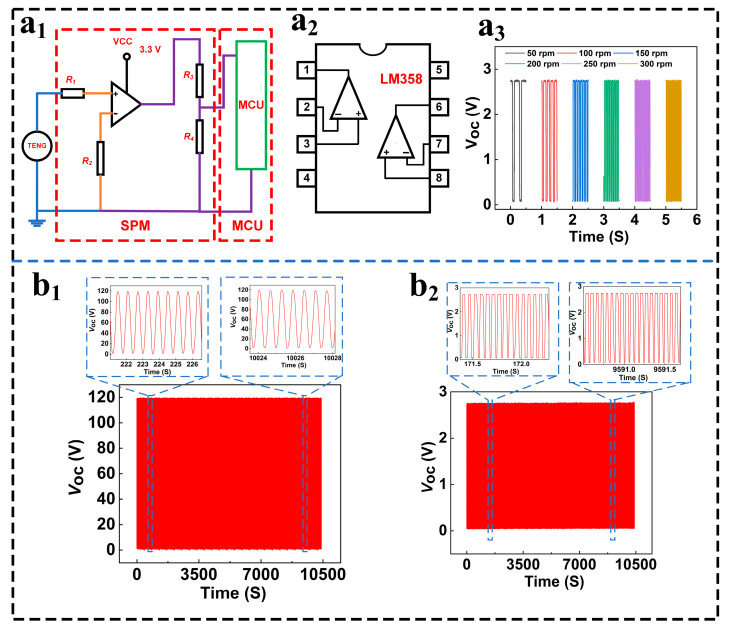
(**a_1_**) Schematic of Signal Processing Module, (**a_2_**) square wave voltage profiles for varied RPM ranges, (**a_3_**) output signals at different rotational speeds and endurance voltage waveform charts featuring MSTKJS-TENG vs. (**b_1_**,**b_2_**) converted square wave signals.

**Figure 5 micromachines-16-01387-f005:**
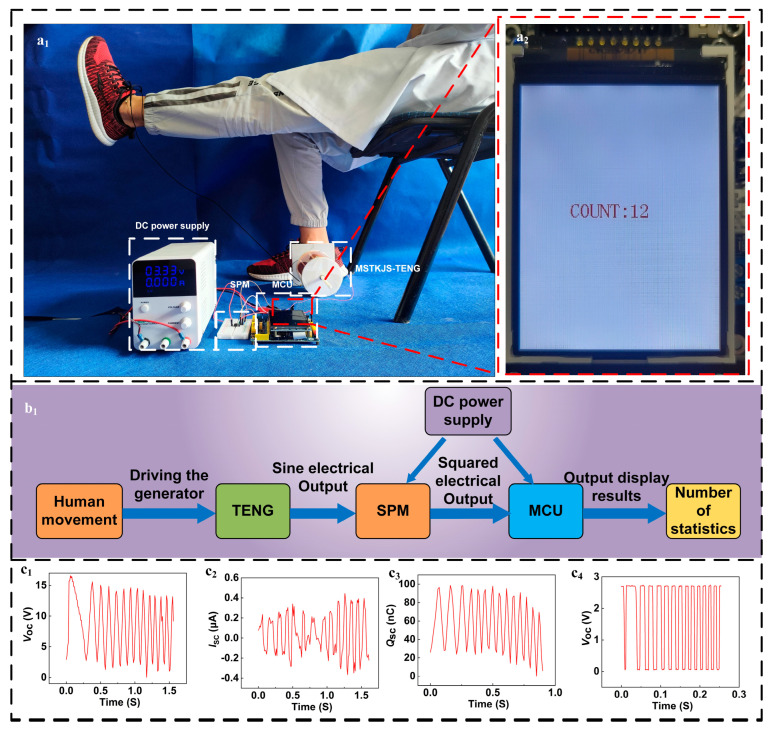
(**a_1_**,**a_2_**) MSTKJS-TENG actual application diagram, (**b_1_**) MSTKJS-TENG principle flowchart, (**c_1_**–**c_3_**) MSTKJS-TENG actual use TENG external output parameters, and (**c_4_**) MSTKJS-TENG actual use TENG external output after Signal Processing Module outputs square wave signal.

## Data Availability

The original contributions presented in this study are included in the article/Appendix A. Further inquiries can be directed to the corresponding author.

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
