# Peer review of "Muscle Strength Training and Monitoring Device Based on Triboelectric Nanogenerator for Knee Joint Surgery"

_micromachines, 2025, doi:10.3390/mi16121387_

Round 1

Reviewer 1 Report

Comments and Suggestions for Authors

The idea of muscle strength training and monitoring signals with TENG is commendable. However, the manuscript would benefit from the following improvements:

1. It would be beneficial to mention other studies applying TENG to human health, such as gait monitoring.

2. Figure 3 is too blurry. Also, some images like b2-b7 are unnecessarily many and could be moved to supplementary information.

3. There are typos: 'Keywrods' should be corrected to 'keywords', and 'HRKJS' should be changed to 'MSTKJS'.

4. The TENG size is large; it should be clarified whether sufficient output can still be achieved if the size is reduced.

5. It would be helpful to add an explanation about how fast the TENG rotates due to torsion when the leg is actually extended.

6. A comparative evaluation of rehabilitation effectiveness, reliability, and monitoring accuracy, compared to expensive conventional total knee arthroplasty (TKA) is necessary.

7. Since rpm varies person to person, it is questionable whether accurate monitoring is possible. How incomplete motions are identified, and how accurate joint angles and forces are evaluated need clarification.

Author Response

Reviewer #1:

The idea of muscle strength training and monitoring signals with TENG is commendable. However, the manuscript would benefit from the following improvements:

Comments 1: It would be beneficial to mention other studies applying TENG to human health, such as gait monitoring.

Response 1: Thank you for your comment. According to the reviewer's comments, we have supplemented relevant studies on TENG applications in human health in the introduction section.

Modification: Top of the Page 2, Line 50. In recent years, TENG has shown great potential in various human health applications. For example, flexible TENG sensors can be used for real-time detection of gait parameters during walking, providing comprehensive gait analysis for patients with lower limb movement disorders

Comments 2: Figure 3 is too blurry. Also, some images like b2-b7 are unnecessarily many and could be moved to supplementary information.

Response 2: Thank you for your comment. We apologize for the blurriness of Figure 3. We have replaced it with a high-resolution version to ensure clear visualization of voltage, current, and power waveforms. Additionally, we have moved Figures 3b3-b7 to Supplementary Information (Figure S3) as suggested, only retaining voltage profiles demonstrating recharge patterns among 50 RPM/Capacitance configurations (Figure 3b2) and a concise summary of charging trends in the main text.

Modification: Bottom of the Page 5, Figure 3.

Comments 3: There are typos: 'Keywrods' should be corrected to 'keywords', and 'HRKJS' should be changed to 'MSTKJS'.

Response 3: We are very sorry for such mistakes. We have corrected “Keywrods” to “Keywords” in the manuscript. Additionally, the incorrect abbreviation “HRKJS” has been uniformly revised to “MSTKJS” throughout the entire manuscript, including text, figures, and captions.

Comments 4: The TENG size is large; it should be clarified whether sufficient output can still be achieved if the size is reduced.

Response 4: Thank you for your valuable comment. The current size of this device (120 mm × 100 mm × 100 mm) which is designed based on the balance of output performance and portability. Compared with traditional instruments, it has achieved a relatively high degree of integration. Since the resistance of this device is provided by internal coil springs and preloading mechanisms, an excessively small device would make it difficult for the coil springs to generate sufficient resistance, failing to meet patients' requirements for resistance training during muscle strength exercises. Moderate miniaturization may be feasible with structural optimization, but excessive reduction compromises performance. We appreciate your suggestion and will explore advanced materials for further size optimization.

Comment 5: It would be helpful to add an explanation about how fast the TENG rotates due to torsion when the leg is actually extended.

Response 5: Thank you for your suggestion. We have supplemented an explanation in section 3.1 Structural design and working principle. This device utilizes a spiral spring to provide torque for the TENG rotor, which is illustrated in Figure 1c. The spiral spring is installed in a spring groove, with one end connected to the central shaft. When the patient pulls the drawstring, the spiral spring drives the rotor to rotate and generates resistance. During the patient’s return movement, the elastically deformed spiral spring reverts to its original state, pulling the drawstring back into the device to complete one pulling cycle.

Modification: Bottom of the Page 3, Line 142. The device uses a spiral spring to provide torque for the TENG rotor. Mounted in a spiral spring groove with one end connected to the central shaft (Figure 1c), the spiral spring drives the rotor to rotate and generate resistance when the patient pulls the drawstring. During the patient’s return movement, the elastically deformed spiral spring reverts to its original state, pulling the drawstring back into the device to complete one pulling cycle.

Comments 6: A comparative evaluation of rehabilitation effectiveness, reliability, and monitoring accuracy, compared to expensive conventional total knee arthroplasty (TKA) is necessary.

Response 6: Thank you for your comment. According to the reviewer's comments, we have supplemented the advantages of MSTKJS-TENG in the abstract. And added a explanation in section 3.4. Demonstration.

Modification: Middle of the Page 1, Line 11. Our team has innovatively applied the TENG self-sensing device to the self-rehabilitation management of knee joint post-surgery. We have developed the "Triboelectric Nanogenerator for Muscle Strength Training of Knee Joint after Surgery" (MSTKJS-TENG), which is significantly more inte-grated than traditional instruments (volume:120 mm × 100 mm × 100 mm) and can real-time track the number and quality of movements completed by patients during muscle strength training. The development of this device has made up for the deficiencies of traditional instruments. It can assist medical staff in remotely evaluating the recovery of patients' postoperative muscle strength to a certain extent, thereby adjusting the training intensity in a timely manner and achieving person-alized guidance.

Comments 7: Since rpm varies person to person, it is questionable whether accurate monitoring is possible. How incomplete motions are identified, and how accurate joint angles and forces are evaluated need clarification.

Response 7: Thank you for raising this critical point. We have supplemented the technical details of motion recognition and parameter evaluation in Section 3.1 Structural design and working principle and 3.4 Demonstration. The MCU algorithm counts and monitors motion completion based on the number of sine signals generated when different patients perform one knee extension, rather than relying on rotational speed values to identify motion cycles. This ensures accurate counting across different users. For incomplete motions, the number of sine signals produced during a knee extension is insufficient and thus not included in the total motion count.

Modification: Bottom of the Page 7, Line 270. Before the training, professionals need to set different counting standards for patients with different limb lengths. The counting standard is determined by the number of square waves generated by the instrument when the patient completes a standard knee extension movement. This number is set as one counting unit (counting as = 1), shown in Figure 5a2.

Reviewer 2 Report

Comments and Suggestions for Authors

This manuscript proposes the MSTKJS-TENG, a triboelectric-nanogenerator–based device for knee-surgery rehabilitation and demonstrates sensing capability for monitoring exercise repetitions. The topic is timely, and the work offers potential impact in intelligent rehabilitation systems. However, several sections require clarification, additional technical detail, and formatting corrections before acceptance.

1. Advantages and limitations of MSTKJS-TENG compared with existing sensors

The Introduction states that MSTKJS-TENG can improve intelligent rehabilitation systems, but the manuscript does not clearly articulate how its advantages compare with existing rehabilitation sensors (e.g., IMUs, load cells, electromyography-based devices).
Please expand the discussion to explicitly address: Unique advantages (self-powered operation, robustness to low-frequency motion, cost, simplicity, etc.), Potential limitations (signal stability, mechanical wear, calibration drift, motion-artifact sensitivity, etc.) This comparison would substantially strengthen the motivation of the study.

2. Section 2.1 “Production of HRKJS-TENG” lacks manufacturing details

The fabrication process is described too briefly.
Please provide more detail, including:

How copper electrodes were patterned and fixed on the acrylic substrate, Fabrication/attachment method of FEP film, Assembly process of the rotor, one-way bearing, and spring mechanism

3. Section 2.2: Meaning of “AO-TENG”

The term AO-TENG appears only once and is not defined anywhere in the manuscript.
Please clarify: Whether AO-TENG is a typographical error, Or a different TENG structure used for calibration

4. Rotary motor type not described

In Section 2.2, “rotary motor (model 142U3E300CACAA165240)” is mentioned without specifying: AC/DC motor, Stepper/servo motor, Torque rating and speed control method

5. Figure formatting issues

The figure set needs reformatting:

 Panels (a), (b), (c), … should appear before the descriptive text below each figure.

 Figure 3 panel labels (b1, b2, …) are not easily readable. Please enlarge or bolden captions within the plots.

6. Additional explanation & structural diagram for the pre-tension device

In Figure 1c, the text states:
“adjust friction between internal wheels driven by pull cords and the pre-tension device by modifying thread length.”
This structure is unclear. Please add:

 A more detailed mechanical sketch or close-up figure

 Explanation of how friction is adjusted

7. Missing pin configuration description for LM358 circuit

In Section 3.3, the explanation references pin #1–#8 of the LM358, but the manuscript does not provide the pin-mapping or circuit diagram showing:

 Which pin corresponds to V+, V–, output, inverting/non-inverting inputs

 Why pins #2/#3 were specifically chosen for TENG signal input
 Please add a clear LM358 pin diagram.

8. Missing text references for Figure 3(b1), 3(b2)

Panels b1 and b2–b7 in Figure 3 are not explicitly mentioned in the main text.
Please add corresponding explanation in the Results section.

9. Terminology consistency (HRKJS-TENG vs. MSTKJS-TENG)

Section 2.1 uses HRKJS-TENG, whereas the main system is described as MSTKJS-TENG throughout the paper.
This inconsistency may be a typographical error and should be corrected for clarity.

Author Response

Reviewer #2:

This manuscript proposes the MSTKJS-TENG, a triboelectric-nanogenerator–based device for knee-surgery rehabilitation and demonstrates sensing capability for monitoring exercise repetitions. The topic is timely, and the work offers potential impact in intelligent rehabilitation systems. However, several sections require clarification, additional technical detail, and formatting corrections before acceptance.

Comments 1: Advantages and limitations of MSTKJS-TENG compared with existing sensors

The Introduction states that MSTKJS-TENG can improve intelligent rehabilitation systems, but the manuscript does not clearly articulate how its advantages compare with existing rehabilitation sensors (e.g., IMUs, load cells, electromyography-based devices).

Please expand the discussion to explicitly address: Unique advantages (self-powered operation, robustness to low-frequency motion, cost, simplicity, etc.), Potential limitations (signal stability, mechanical wear, calibration drift, motion-artifact sensitivity, etc.) This comparison would substantially strengthen the motivation of the study.

Response 1: Thank you for your comment. Based on the reviewers' suggestions, we have added references (Energy Environ. Sci. 2015, 8, 2250-2282, ACS Nano. 2013, 7, 9533-9557) regarding the advantages and limitations of TENG in the introduction section.

Modification: Top of the Page 2, Line 46. The Triboelectric Nanogenerator (TENG) is an energy harvesting device based on contact electrification and electrostatic induction coupling effects invented by Wang Zhong Lin's team in 2012. It is demonstrated remarkable characteristics in self-sensing applications due to its low cost, light weight, and high energy harvesting efficiency.

Comments 2: Section 2.1 “Production of HRKJS-TENG” lacks manufacturing details

The fabrication process is described too briefly. Please provide more detail, including:

How copper electrodes were patterned and fixed on the acrylic substrate, Fabrication/attachment method of FEP film, Assembly process of the rotor, one-way bearing, and spring mechanism

Response 2: Thank you for your comment. I'm very sorry HRKJS-TENG should actually be MSTKJS-TENG. According to the reviewer's comments, we have supplemented the details of manufacturing in section 2.1 Production of MSTKJS-TENG.

Modification: Middle of the Page 2, Line 73. The overall structure of MSTKJS-TENG is drawn using SolidWorks software and processed by a 3D printer. The main shell material is acrylic board, which is cut by a laser cutting machine. The external dimensions are 120 (length) mm × 100 (width) mm × 100 (height) mm. The copper electrodes evenly distributed on the acrylic board shell have a thickness of 1 mm, and each electrode is of equal size, which is conducive to the stable output of signals. The copper electrodes are adhered to the acrylic board with epoxy resin adhesive. The thickness of the polytetrafluoroethylene (FEP) film is 0.08 mm, and it is combined with the rotor through epoxy resin adhesive. The size of the one-way bearing is 8 (inner diameter) mm × 12 (outer diameter) mm × 12 (height). It is placed at the center of the teng rotor and connected to the puller. The internal structure of the puller consists of a spring, a rotor wheel, and a central shaft. Additionally, as shown in Figure 1b1, this device is equipped with a preload structure for adjusting the motion difficulty, which is composed of a threaded rod (M4 × 50 mm), a spring (outer diameter 8 mm, free length 20 mm), and a friction plate (made of polytetrafluoroethylene, diameter 10 mm).

Comments 3: Section 2.2: Meaning of “AO-TENG”

The term AO-TENG appears only once and is not defined anywhere in the manuscript.

Please clarify: Whether AO-TENG is a typographical error, Or a different TENG structure used for calibration

Response 3: Thank you for your comment. We are very sorry for such mistakes. In the revised text, we have changed AO-TENG to MSTKJS-TENG.

Comments 4: Rotary motor type not described

In Section 2.2, “rotary motor (model 142U3E300CACAA165240)” is mentioned without specifying: AC/DC motor, Stepper/servo motor, Torque rating and speed control method

Response 4: Thank you for your comment. Based on the reviewers' suggestions, we have added the relevant parameters of the motor in section 2.2 Experimental equipment.

Modification: Bottom of the Page 2, Line 91. The equipment used to measure and acquire signals from MSTKJS-TENG includes a programmable electrometer (model 6514, Keithley, USA), a data acquisition card (USB-6356 BNC, IN, USA), and a rotary motor (model 142U3E300CACAA165240) providing external excitation. Signals collected during the experiment were all recorded using LabVIEW 2019 software.

Comments 5: Figure formatting issues

The figure set needs reformatting:

Panels (a), (b), (c), … should appear before the descriptive text below each figure. Figure 3 panel labels (b1, b2, …) are not easily readable. Please enlarge or bolden captions within the plots.

Response 5: Thank you for your comment. According to your suggestion, we have made modifications to the script.

Comments 6: Additional explanation & structural diagram for the pre-tension device

In Figure 1c, the text states:

adjust friction between internal wheels driven by pull cords and the pre-tension device by modifying thread length.”

This structure is unclear. Please add:

A more detailed mechanical sketch or close-up figure

Explanation of how friction is adjusted

Response 6: Thank you for your comment. According to the experts' suggestions, we have added the description and structural diagram of the pre-tensioning device in section 3.1 Structural design and working principle.

Modification: Top of the Page 3, Line 103. The overall structure and pre-tightening structure of MSTKJS-TENG are shown in Figures 1b1 and Figures S2 respectively. This device is equipped with a pre-tightening structure for adjusting the motion difficulty. This structure consists of a threaded rod, a spring, and a friction plate. By rotating the threaded rod, the friction plate and the spring form an extrusion force, thereby increasing the resistance of the spring during its elastic deformation, and thus changing the resistance during knee extension. When performing the "extension" action, the pulling rope drives the spring to undergo elastic deformation, which in turn drives the rotor to rotate, thereby providing torque to the TENG rotor. When completing the "flexion" action, the spring returns to its original state and pulls the rope back into the pull disc. Since the TENG rotor uses a one-way bearing, when the pull disc returns to its deformed state, the TENG will not be affected by the counterforce.

Comment 7: Missing pin configuration description for LM358 circuit

In Section 3.3, the explanation references pin #1–#8 of the LM358, but the manuscript does not provide the pin-mapping or circuit diagram showing:

Which pin corresponds to V+, V–, output, inverting/non-inverting inputs

Why pins #2/#3 were specifically chosen for TENG signal input

Please add a clear LM358 pin diagram.

Response 7: Thank you for your comment. We add a clear LM358 pin diagram,in Figure 4, according to the reviewers suggestion.

Comment 8: Missing text references for Figure 3(b1), 3(b2)

Panels b1 and b2–b7 in Figure 3 are not explicitly mentioned in the main text.

Please add corresponding explanation in the Results section.

Response 8: Thank you for your comment. According to the reviewer's comments, We have provided explanations and descriptions in section 3.2. Power generation performance test

Modification: Bottom of the Page 6, Line 230. As shown in Figure b2. Under the condition where the rotational speed is stabilized at 50 rpm, when all the test capacitors aim to charge to 10 V, the charging rates of different capacity capacitors show significant differences. Among them, the 0.1 μF capacitor has the fastest charging rate, requiring only 1 second to complete the charging process from the initial state to 10 V; the capacitor with a capacity of 0.22 μF has a charging efficiency second-best, reaching 10V voltage in only 5 seconds; the capacitor with a capacity of 1.0 μF has a further increased charging time, taking 10 seconds to charge up to 10 V; while the capacitor with a capacity of 2.2 μF has the slowest charging rate, from the start of charging to the voltage reaching 10 V, the entire process takes 50 seconds. In the subsequent experiments, different capacitors were replaced for multiple sets of experiments, and the output performance was stable (Figure S3).

Comment 9: Terminology consistency (HRKJS-TENG vs. MSTKJS-TENG)

Section 2.1 uses HRKJS-TENG, whereas the main system is described as MSTKJS-TENG throughout the paper.

This inconsistency may be a typographical error and should be corrected for clarity.

Response 9: We are very sorry for such mistakes. In the revised text, we have modified it.

Reviewer 3 Report

Comments and Suggestions for Authors

In this work the authors investigate a triboelectric nanogenerator self-powered sensor based system for rehabilitation post knee joint surgery. They describe the sensor operation and experimentally study the sensor output vs input speed. Then they describe how the sensor output is digitized and the exercise movement is counted using MCU.

The research idea is very interesting but the content is lacking and can be significantly improved. Here are the reviewer's comments.

1. Please provide details (figure and explanation) of the pre-tension structure for exercise difficulty adjustment. This is a key part of the device not explored.
2. Please provide mechanical theoretical analysis involving input force, wind spring, and pre-tension structure. The authors can refer to theoretical analysis shown in the paper: [https://doi.org/10.1016/j.bios.2025.117995] where similar cable structure is used with wind spring, TENG, and exercise difficulty adjustment.
3. Please provide details of MCU algorithm. How many TENG peaks correspond to one exercise count?
4. Section 2.1 incorrectly mentions use of PTFE, when actually FEP is used in the paper.
5. The supplementary content is not referred to in the manuscript.
6. Based on Figure 3a4 the peak power is observed at ~10^7ohms. However in Section 3.2 it is incorrectly mentioned optimal power will be at 10^8ohms.
7. In Figure 5b1 there should be arrow between SPM and MCU.

Comments on the Quality of English Language

Some of the English in the paper is wrong. Such as in Section 3.2, use of the word "telecommunication signals", "short circuit peak-to-peak voltage", "increasing wave numbers", etc.

Author Response

Reviewer #3:

In this work the authors investigate a triboelectric nanogenerator self-powered sensor based system for rehabilitation post knee joint surgery. They describe the sensor operation and experimentally study the sensor output vs input speed. Then they describe how the sensor output is digitized and the exercise movement is counted using MCU.

The research idea is very interesting but the content is lacking and can be significantly improved. Here are the reviewer's comments.

Comments 1: Please provide details (figure and explanation) of the pre-tension structure for exercise difficulty adjustment. This is a key part of the device not explored.

Response 1: Thank you for emphasizing this key component. We have supplemented detailed information about the pre-tension structure in Section 3.1 Structural design and working principle, and added a dedicated in Figure S2.

Modification: Top of the Page 3, Line 103. The overall structure and pre-tightening structure of MSTKJS-TENG are shown in Figures 1b1 and Figure S2 respectively. This device is equipped with a pre-tightening structure for adjusting the motion difficulty. This structure consists of a threaded rod, a spring, and a friction plate. By rotating the threaded rod, the friction plate and the spring form an extrusion force, thereby increasing the resistance of the spring during its elastic deformation, and thus changing the resistance during knee extension. When performing the "extension" action, the pulling rope drives the spring to undergo elastic deformation, which in turn drives the rotor to rotate, thereby providing torque to the TENG rotor. When completing the "flexion" action, the spring returns to its original state and pulls the rope back into the pull disc. Since the TENG rotor uses a one-way bearing, when the pull disc returns to its deformed state, the TENG will not be affected by the counterforce.

Comments 2: Please provide mechanical theoretical analysis involving input force, wind spring, and pre-tension structure. The authors can refer to theoretical analysis shown in the paper: [https://doi.org/10.1016/j.bios.2025.117995] where similar cable structure is used with wind spring, TENG, and exercise difficulty adjustment.

Response 2: We appreciate your guidance on relevant literature. We have added a mechanical theoretical analysis section 3.1 Structural design and working principle in the revised manuscript, referring to the method in the cited paper.

Modification: Middle of the Page 3, Line 115. The torsion spring provides a restoring torque approximately proportional to its angle of rotation, but its operation in a pull plate is closer to a constant force. For a classic planar torsion spring, its torque can be approximately expressed as:

Comments 3: Please provide details of MCU algorithm. How many TENG peaks correspond to one exercise count?

Response 3: Thank you for raising this critical point. We have supplemented the technical details of motion recognition and parameter evaluation in section 3.1 Structural design and working principle.

Modification: Bottom of the Page 4, Line 155. Since the MSTKJS-TENG is equipped with 5 pairs of electrodes and 5 FEP film paddles, it generates 5 cycles of sine waves after rotating one full circle. However, the backend MCU module responsible for counting cannot identify sine wave signals. Therefore, the electrical signals generated by the MSTKJS-TENG need to be transmitted to the backend SPM for processing via wires to convert them into square waves. Finally, the square waves are recognized and processed by the backend MCU to achieve the counting function (Detailed information about the MCU algorithm can be found in the Supplementary Information).

Comments 4: Section 2.1 incorrectly mentions use of PTFE, when actually FEP is used in the paper.

Response 4: We apologize for this inconsistency. We have corrected “PTFE” to “FEP” in section 2.1 Production of MSTKJS-TENG to align with the material used in the experiment and other sections of the manuscript.

Comments 5: The supplementary content is not referred to in the manuscript.

Response 5: Thank you for your comments. We have revised the manuscript to explicitly reference supplementary content at relevant positions. All supplementary figures are numbered sequentially and cross-referenced in the main text.

Comments 6: Based on Figure 3a4 the peak power is observed at~10^7ohms. However in Section 3.2 it is incorrectly mentioned optimal power will be at 10^8ohms.

Response 6: We apologize for this calculation error. We have corrected the optimal load resistance value in Section 3.2 from 108 ohms to 107ohms, which is consistent with the data in Figure 3a4.

Comments 7: In Figure 5b1 there should be arrow between SPM and MCU.

Response 7: Thank you for noticing this detail. We have revised Figure 5b1 by adding a directional arrow between SPM and MCU, indicating the transmission of square wave signals from the signal processing module to the microprogrammed control unit.

Round 2

Reviewer 3 Report

Comments and Suggestions for Authors

In this work the authors investigate a triboelectric nanogenerator self-powered sensor based system for rehabilitation post knee joint surgery. They describe the sensor operation and experimentally study the sensor output vs input speed. Then they describe how the sensor output is digitized and the exercise movement is counted using MCU.

The authors have made the requested changes to the manuscript, however the reviewer has some additional comments as below,

1. It seems the torque balance equation is simply T_human = T_spring + T_friction-plate. So how is the resistive force increased? Is it increased by T_spring=theta_spring×k_spring, where k_spring is spring stiffness and theta_spring is number of spring rotations?

2. In line 219,223 Equation 5,6 is mistakenly written as Equation 1,2.

3. In Figure 5b, between TENG and SPM should be sine electrical output, and between SPM and MCU should be squared electrical output, above the arrows.

4. FEP is Fluorinated ethylene propylene and is incorrectly described in Section 2.1

Author Response

In this work the authors investigate a triboelectric nanogenerator self-powered sensor based system for rehabilitation post knee joint surgery. They describe the sensor operation and experimentally study the sensor output vs input speed. Then they describe how the sensor output is digitized and the exercise movement is counted using MCU.

The authors have made the requested changes to the manuscript, however the reviewer has some additional comments as below.

Comments 1: It seems the torque balance equation is simply T_human = T_spring + T_friction-plate. So how is the resistive force increased? Is it increased by T_spring=theta_spring×k_spring, where k_spring is spring stiffness and theta_spring is number of spring rotations?

Response 1: Thank you for your valuable question. As derived from the torque balance equation, the relationship Thuman body = Tspring + Tfriction-plate holds, where Tspring = θspring × kspring. Since the spring integrated into the pull plate of the device is a fixed component, its stiffness kspring is an inherent, unchangeable parameter. In contrast, the frictional torque follows the relationship Tfriction-plate = θfriction-plate × kfriction-plate. here, θfriction-plate denotes the surface roughness of the friction plate (a fixed characteristic of the plate material), while kfriction-plate represents the extrusion force between the preload mechanism and the friction plate-a variable parameter that can be adjusted. Therefore, the adjustment of the device’s resistive force is primarily achieved by regulating Tfriction-plate through modifying kfriction-plate (i.e., adjusting the extrusion force via the preload mechanism), rather than relying on changes to the fixed Tspring.

Comments 2: In line 219, 223 Equation 5, 6 is mistakenly written as Equation 1, 2.

Response 2: We are very sorry for such mistakes. We have carefully checked the manuscript and corrected the equation numbering. Specifically, the equations previously incorrectly labeled as Equation (1) and Equation (2) in these lines have been revised to Equation (5) and Equation (6) respectively, which is consistent with the equation numbering sequence in the manuscript.

Comments 3: In Figure 5b, between TENG and SPM should be sine electrical output, and between SPM and MCU should be squared electrical output, above the arrows.

Response 3: Thank you for your comment. We have revised Figure 5b according to your suggestion. Specifically, we have added labels “Sine electrical output” above the arrow between TENG and SPM, and “Squared electrical output” above the arrow between SPM and MCU in the figure.

Comments 4: FEP is Fluorinated ethylene propylene and is incorrectly described in Section 2.1

Response 4: We are very sorry for such mistakes. In the revised text, we have modified it.
